# Investigating the Joint Effect of Allostatic Load among Lesbian, Gay, and Bisexual Adults with Risk of Cancer Mortality

**DOI:** 10.3390/ijerph20126120

**Published:** 2023-06-13

**Authors:** Justin Xavier Moore, Sydney Elizabeth Andrzejak, Tracy Casanova, Marvin E. Langston, Søren Estvold, Prajakta Adsul

**Affiliations:** 1Cancer Prevention, Control & Population Health Program, Department of Medicine, Augusta University, Augusta, GA 30912, USA; 2Institute of Preventive and Public Health, Medical College of Georgia, Augusta University, Augusta, GA 30912, USA; 3Department of Psychiatry and Health Behavior, Georgia Cancer Center, Augusta University, Augusta, GA 30912, USA; 4Department of Epidemiology and Population Health, Stanford University, Stanford, CA 49305, USA; 5Department of Family Medicine, Augusta University, Augusta, GA 20912, USA; 6Division of Epidemiology, Biostatistics and Preventive Medicine, University of New Mexico, Albuquerque, NM 87131, USA; 7Cancer Control and Population Sciences Research Program, Comprehensive Cancer Center, University of New Mexico, Albuquerque, NM 87131, USA

**Keywords:** life-course, cumulative stress, psychosocial stress, sexual and gender minority health, disparities, minority stress, allostatic load

## Abstract

Sexual minorities (SM) have higher chronic physiologic stress as indicated by allostatic load (AL), which may be explained in part by consistent experiences of discriminatory practices. This is one of the first studies to examine the joint effects of SM status and AL on the association with long-term risk for cancer death. Retrospective analyses were conducted on 12,470 participants using National Health and Nutrition Examination Survey (NHANES) from years 2001 through 2010 linked with the National Death Index through December 31, 2019. Cox proportional hazards models estimated adjusted hazard ratios (aHRs) of cancer deaths between groups of SM (those reporting as gay, lesbian, bisexual, or having same-sex sexual partners) status and AL. SM adults living with high AL (*n* = 326) had a 2-fold increased risk of cancer death (aHR: 2.55, 95% CI: 1.40–4.65) when compared to straight/heterosexual adults living with low AL (*n* = 6674). Among those living with high AL, SM (*n* = 326) had a 2-fold increased risk of cancer death (aHR: 2.26, 95% CI: 1.33–3.84) when compared to straight/heterosexual adults with high AL (*n* = 4957). SM with high AL have an increased risk of cancer mortality. These findings highlight important implications for promoting a focused agenda on cancer prevention with strategies that reduce chronic stress for SM adults.

## 1. Introduction

Coined in the mid-1990s, McEwen and Seeman theorized the concept of allostatic load (AL) and “weathering” as the physiological effects of life-course stress, or the cumulative ‘wear and tear’ on the body, from repeated exogenous stressors [1,2,3,4]. AL assesses biomarkers from multiple organ systems in an attempt to quantify the biological effects of chronic stress and the over-activation of several adaptive processes. AL has been used to predict morbidity and mortality of various diseases, including diabetes mellitus [5], cardiovascular disease [6], and particularly cancer [7]. Sexual minorities (SM) have unique barriers to healthcare, the sexual minority stress model proposed that discrimination and stigmatization of minority sexual orientations contribute to later health disparities [8,9]. Mays et al. observed that sexual orientation was associated with chronic stress, or AL, among adult men [10]. They observed that gay men had significantly lower levels of AL, while bisexual men had significantly higher levels of AL when compared to exclusively heterosexual men [10]. 

Marginalized communities, racial/ethnic minorities, and sexual minorities bear the disproportional burden of discriminatory practices and resulting stressors. Stigmatization, structural barriers, and medical mistrust are additional psychosocial stressors that SM navigate when seeking healthcare [11,12,13,14]. Stigmatized experiences within the healthcare setting, including lack of cultural competency, misgendering and deadnaming, heteronormative assumptions, discrimination, and prejudice, as well as inadequate sexual minority healthcare services, contribute to an environment of hostility and intolerance. This environment, in turn, results in higher levels of chronic stress experienced by sexual minorities. In a recent self-report study of 2534 SM individuals, 30% reported having a previous diagnosis of cancer and approximately 20–30% reported not receiving the recommended cancer screenings [15]. Moreover, Langston et al. observed that lesbian, gay, and bisexual adults are more than 3-fold more likely to look for information about cancer health or medical topics, and 2.3 more likely to worry about developing cancer, but are 83% less likely to seek or discuss health information from a physician when compared to heterosexual adults [16]. 

To date, there is a paucity of research that has examined the relationship between sexual minorities and the risk of cancer death, and more specifically the moderating role of chronic physiologic stress due to persistent psychosocial stressors (e.g., derived from social environmental influencers such as bigotry, homophobia/biphobia, dehumanization, and discrimination) among a current, nationally representative sample of United States (US) adults. In this study, we examined whether AL modified the association between self-identifying sexual minorities with cancer mortality risk in US adults. 

## 2. Materials and Methods

### 2.1. Study Design and Participants

A retrospective cohort analysis was performed using data from National Health and Nutrition Examination Survey (NHANES). NHANES can be linked with the National Center for Health Statistics (NCHS) 2019 National Death Index (NDI) file and is a representative sample of non-institutionalized US residents. The NHANES program oversamples participants aged 60 and older, Hispanic and non-Hispanic Black (NH-Black). NHANES surveys demographic, socioeconomic, dietary, and health-related questionnaires, self-reported medication use for health conditions, and clinical measurements of blood pressure, blood glucose, triglycerides, and HDL cholesterol. The association of AL with SM and cancer mortality was examined using NHANES survey participant data from 2001 through 2010 linked with NDI follow-up data through 31 December 2019. Analysis was performed among NHANES participants with data on biomarkers. Participants who were missing biomarkers for AL, follow-up time (i.e., information regarding censoring or death after NHANES linkages with the NDI), information regarding their sexual orientation status, were under the age of 18, or were currently pregnant were excluded. This analysis included all participants aged 18 and older; corresponding to a total of 12,470 over the 19-year study period for the main analysis (Figure 1). Mortality status or vital status for participants was determined through NHANES-NDI linked file. 

### 2.2. Ethical Statement

The Augusta University Institutional Review Board considered this study exempt from review because of the use of publicly available, de-identified data.

### 2.3. Sexual Minority Status

Beginning in 2001, the NHANES interview questioned on both sexual orientation identity (e.g., heterosexual, lesbian/gay, bisexual) and the sex of sexual partners since age 18 and in the year prior to the interview. NHANES investigators surveyed participants regarding their sexual orientation with the following question: “Do you think of yourself as heterosexual or straight (that is, sexually attracted only to [opposite sex] women/men); homosexual or gay (that is sexually attracted only to [same sex] men/women); bisexual (that is, sexually attracted to men and women); something else; or you’re not sure?” NHANES investigators additionally queried on the number of same sex partners had in a lifetime: “In your lifetime, with how many women/men have you had sex?” Questions on sexual behavior and orientation were captured among NHANES participants aged 20 through 59 years. We followed the Mays et al. categorization for SM status, participants were categorized as follows: (1) those reporting a lesbian or gay identity, regardless of sexual history (*n* = 192); (2) those reporting a bisexual identity, regardless of sexual history (*n* = 289); (3) those indicating positive lifetime histories of same-sex sexual partners (men who have sex with men (MSM) and women who have sex with women (WSW); *n* = 358) in the absence of a current lesbian, gay or bisexual identity; or (4) straight/heterosexual (*n* = 11,631) including those who explicitly self-identified as heterosexual or reported no same-sex sexual partners or gay/bisexual identity. For statistical comparisons, we used the straight/heterosexual group as the reference population. We did not include participants who reported “something else”, “not sure”, “don’t know”, or “refused”, as these groups represent other sexual identities outside the scope of the current analysis [10]. 

### 2.4. Allostatic Load Definition

Allostatic load (AL) has been defined using varying components, although most incorporate biomarker measures from three different categories including cardiovascular, metabolic, and immune systems [17]. The study at present elected to define AL using the Geronimus et al. (2006) and Moore et al. (2021) taxonomies [18,19], as there is no census definition. Components of AL included body mass index (BMI), diastolic blood pressure (DBP), glycohemoglobin (hemoglobin A1c), systolic blood pressure (SBP), total cholesterol, serum triglycerides, serum albumin, serum creatinine, and C-reactive protein (CRP). Gender was considered by NHANES as (1) male, (2) female, and (3) missing. High-risk thresholds for each AL component were determined by examining the *sex reported at survey* specific distributions of each component among the entire study sample with complete biomarker data. High-risk thresholds were determined by either being above the 75th percentile for BMI, CRP, DBP, glycated hemoglobin, SBP, total cholesterol, serum triglycerides, and serum creatinine [20,21]; or below the 25th percentile for serum albumin. Therefore, each NHANES participant was scored as either 1 (high-risk) or 0 (low risk) based on sex at baseline survey-specific cutoffs for each component (Appendix A). The total AL score was calculated by summing the individual components, and this score ranged from 0 to 9. Participants were further categorized with an AL score greater or equal to 3 as having high AL [10,17]. 

### 2.5. Allostatic Load and Sexual Orientation

After categorizing NHANES participants based on the distribution of AL components and their self-reported sexual orientation status, we created a variable examining the intersection of AL and SM status. This variable was categorized into eight groups; (1) straight/heterosexual living with low AL (*n* = 6674), (2) straight/heterosexual living with high AL (*n* = 4957), (3) gay/lesbian living with low AL (*n* = 128), (4) gay/lesbian living with high AL (*n* = 64), (5) bisexual living with low AL (*n* = 177), (6) bisexual living with high AL (*n* = 112), (7) MSM and WSW living with low AL (*n* = 208), and (8) MSM and WSW living with high AL (*n* = 150). Our original interest was to present sexual orientation/sexual behavior-specific outcomes in the above-mentioned eight groups. However, given the limited sample size, we concatenated sexual behavior and sexual minority groups into a dichotomous variable (e.g., combined gay/lesbian, bisexual, MSM, WSW vs. heterosexual/straight) with allostatic load status (high vs. low). Thus, we present our primary findings in four groups: (1) straight/heterosexual with low allostatic load, (2) straight/heterosexual with high allostatic load, (3) Gay, Lesbian, Bisexual, MSM, WSW with low allostatic load, and (4) Gay, Lesbian, Bisexual, MSM, WSW with high allostatic load. However, we present more granular findings by the eight groups of sexual orientation and allostatic load within supplemental results.

### 2.6. Primary Outcome of Interest, Cancer Death

Our primary outcome of interest was time to cancer-related death. Follow-up data for this analysis was available through 31 December 2019, based on NDI-NHANES publicly available linkages. The primary determination of mortality for eligible NHANES participants is based upon matching survey records to the NDI, although additional redundant sources are also incorporated, including the Social Security Administration, the Centers for Medicare and Medicaid Services, data collection, NCHS’ follow-up surveys (e.g., NHEFS), and ascertainment of death certificates. 

### 2.7. Participant Characteristics

The sociodemographic factors from the NHANES survey included age, sex, race/ethnicity, education, annual household income, smoking status, and alcohol consumption. Race and ethnicity were categorized as non-Hispanic (NH)-Black, NH-White, Hispanic (including other Hispanic and Mexican American), other and mixed race. Sex was categorized as self-reported (1) male; (2) female; (3) missing. The NHANES educational attainment variable was categorized into (1) less than high school education; (2) high school graduate/GED/or equivalent; (3) some college; (4) college graduate or above and (5) unknown/refused to answer. Annual household income in a range of dollars was given to survey respondents as a categorical multi-level question, and we dichotomized participants as (1) earning less than $20,000 per year or (2) more than $20,000 per year. Marital status was categorized as (1) married or living with a partner; (2) widowed, divorced, or separated; (3) never married; (4) missing, do not know, refused to answer. Participants that had not smoked 100 cigarettes in their lifetime were categorized as never smokers, while participants with at least 100 cigarettes smoked in their lifetime but no current smoke use were categorized as past smokers. Participants with at least 100 lifetime cigarettes used and current smoking use were categorized as current smokers [22]. NHANES questioned whether participants drank any alcoholic beverages such as liquor, beer, wine, wine coolers, and any other type of alcoholic beverages in any one year and had at least 12 drinks of any type of alcoholic beverage. Further, survey interviewers clarified “By a drink, I mean a 12 oz. beer, a 5 oz. glass of wine, or one and half ounces of liquor.” In addition, NHANES interviewers questioned, “In the past 12 months, on those days that you drank alcoholic beverages, on the average, how many drinks did you have?” As per the classification provided by the National Institute on Alcohol Abuse and Alcoholism, we categorized alcohol consumption into moderate levels (one drink per day for women or two drinks per day for men) and heavy alcohol use (more than one drink per day for women and more than two drinks per day for men) [23]. From these, we categorized participants as either (1) never drinkers (not having more than 12 drinks during lifetime), (2) former drinkers (having at least 12 drinks during lifetime but not in the past year), (3) moderate drinkers (up to one drink per day for women, and up to two drinks per day for men), and (4) heavy drinker (2+ drink per day for women, 3+ drinks per day for men) [23].

### 2.8. Statistical Analysis

Analyses were performed for descriptive statistics (i.e., relative frequencies and proportions for categorical variables, and means and standard errors for continuous variables) using NHANES-generated sampling statistical strata, clusters, and weights as designated and described in detail within the NHANES methodology handbook [24]. NHANES only measures biomarkers among a random sample of participants each survey period, and in turn, created subsample weights to account for the probability of being selected into the subsample component and additional non-response bias. Categorical variables were presented as weighted percentages and continuous variables as mean and associated 95% confidence intervals (CIs). We compared characteristics by categories of the intersectionality of AL and sexual minority status using Rao–Scott Chi-Square tests for categorical variables and weighted Wald F-tests for continuous variables. We estimated the mean survival times using the product-limit method of the Kaplan–Meier survival estimator. We examined the survival function of cancer mortality by AL status overall, and then stratified by groups of AL/SM status using the Kaplan–Meier method. The proportionality assumption was assessed for the primary exposure variable (intersectionality of AL and SM status) by examining the proportion of 1000 simulations that contain maximum cumulative martingale residuals larger than the observed maximum cumulative residuals using the SAS procedure ‘supremum test’. None of the levels of the exposure had *p*-values that were statistically significant (*p*-value < 0.05), and therefore, none of the residuals were larger than expected and the proportional hazards assumptions were not rejected [25,26]. Thus, survey-weighted Cox proportional hazards models were performed to attain hazard ratios (HRs) and associated 95% confidence intervals (CIs) comparing AL/sexual minority groups’ risks for cancer-related death. For all time-to-event analyses (including competing risks analyses below), NHANES participants contributed to follow-up time starting from their baseline interview, and participants were censored at the time of their event, death, or end of follow-up (31 December 2019). We adjusted our models for (1) age and (2) sociodemographics (sex reported at the survey, race, income, education, smoking status, and alcohol consumption). Confounders were selected based on factors available within NHANES, biological rationale, and finally, whether significance was observed in bivariate analysis. We examined the multiplicative interactions of AL with sexual minority status by introducing an interaction term within our model and presenting the corresponding *p*-value for this association.

In secondary analyses (presented in Appendix A), a time-to-event analysis was conducted to account for competing risks. For competing risks analyses we treated our analytic cohort as a simple random sample and conducted un-weighted analyses. We conducted Fine and Gray Cox proportional hazard models [27] to examine all-cause mortality as a potential competing risk for cancer deaths and presented results from our competing risks analysis as sub-distribution hazard ratios (SHR) and associated 95% CIs. Lastly, we estimated the cumulative incidence function (CIF) using competing risks analysis by categories of AL with sexual orientation and present the age-adjusted CIF plot. All statistical analyses were performed using SAS (version 9.4, SAS Institute, Inc., Cary, NC, USA) and Stata (version 17, StataCorp, 4905 Lakeway Drive College Station, TX 77845, USA). 

## 3. Results

### 3.1. Descriptive Characteristics by Sexual Orientation/Allostatic Load

In Table 1 we present the demographic characteristics of the NHANES participant sample (*n* = 12,470, weighted *n* = 46,347,598, Figure 1) during their baseline interview by sexual minority status stratified by those with high and low AL. We observed no statistically significant differences between allostatic load scores by sexual minority groups (AL mean scores: gay/lesbian = 1.66, SE = 0.12; bisexual = 1.92, SE = 0.12; homosexually experienced = 1.85, SE = 0.12; heterosexual/straight = 1.90, SE = 0.03) (Data not shown). Gay/lesbian adults living with high AL were: less likely to report female sex at the survey (38.3% vs. 46.7%), be older in age (mean age 42.4 vs. 36.5), more likely to identify as NH-Black (12.5% vs. 7.9%), less likely to be married or live with a partner (28.2% vs. 64.8%), more likely to have some college or associate’s degree educational attainment (41.4% vs. 31.7%), more likely to have health insurance (83.0% vs. 76.3) and go to 1 or more healthcare visits in a year (80.5% vs. 78.8%), more likely to be current smokers (34.8% vs. 26.0%), have a prior heart attack (2.6% vs. 0.8%), and more likely be HIV positive (4.9% vs. 0.1%) when compared with straight/heterosexual adults living with low AL. Bisexual adults living with high AL (*n* = 112, weighted *n* = 1,049,553) were: more likely to report female sex at the survey (68.9%), identify as NH-Black (14.5%), have some college or associate’s degree educational attainment (39.6%), be current smokers (39.5%), have a prior heart attack (2.1%), and had a nearly 5-fold higher prevalence of depressive disorder (22.3% vs. 5.4%) when compared with straight/heterosexual adults living with low AL. Adults that reported MSM and WSW with high AL were more likely to report female sex at the survey (72%), be older in age (mean age 43.5 years), identify as NH-black (14.9%), be current smokers (41.2%), ever had a heart attack (16.3% vs. 8.3%), and have depressive disorder (17.4%) when compared with straight/heterosexual adults living with low AL.

### 3.2. Association between Sexual Orientation/Allostatic Load with Risk of Cancer Death

In survey-weighted Cox Proportional Hazard models, there were 222 (weighted *n* = 1,955,041) deaths attributed to cancer among our analytic sample (Table 2, Figure 2). Among sexual minorities (SM: gay, lesbian, bisexual, men who have sex with men (MSM), and women who have sex with women (WSW)) living with high AL there was a 3-fold increased risk of cancer death (age-adjusted HR: 3.32, 95% CI: 1.86–5.93) when compared to straight/heterosexual adults with low AL. After adjustments for age and sociodemographic factors, sexual minorities with high AL (model 2 HR: 2.55, 95% CI: 1.33–4.65) were at an increased risk of cancer death. When stratifying analysis among those with high AL status, sexual minorities continued to have an increased risk of cancer mortality (model 2 HR: 2.26, 95% CI: 1.33–3.84). 

In Appendix A we elucidate cancer risk specifically among gay, lesbian, bisexual, MSM, and WSW. Gay/lesbian adults with high AL (7.5% vs. 1.1%; age-adjusted HR: 4.94, 95% CI: 1.99–12.27), and MSM and WSW adults living with high AL (4.9% vs. 1.1%; age-adjusted HR: 3.05, 95% CI: 1.62–5.74) were all at increased risks of cancer death when compared to straight/heterosexual adults living with low AL. After adjustment for age, reported sex at survey, race, income, education, smoking status, and alcohol consumption gay/lesbian adults living with high AL (model 2 HR: 4.17, 95% CI: 1.78–9.79) and MSM and WSW adults living with high AL (model 2 HR: 2.07, 95% CI: 1.09–3.90) when compared to straight/heterosexual adults living with low AL.

### 3.3. Sensitivity Analyses—Fine and Gray Competing Risks Analysis

In our Fine and Gray Cox proportional hazard models there were an additional 484 deaths from other causes among our sample (Appendix A). Results were similar to the weighted Cox proportional hazard model yet slightly attenuated in magnitude. Most importantly, gay/lesbian adults living with high AL were between 3.7-fold (age-adjusted HR: 3.74, 95% CI: 1.39–10.07) and 3.03-fold (model 2 HR: 3.03, 95% CI: 1.11–8.26) increased risk of dying from cancer when compared to straight/heterosexual adults with low AL (Appendix A).

## 4. Discussion

The current study is one of the few studies examining the joint effects of sexual minorities and AL on the association with long-term risk for cancer death. We observed that in fully adjusted models, SM (i.e., gay, lesbian, bisexual, MSM, and WSW) with high AL had a 2-fold increased risk of cancer death when compared to straight/heterosexual individuals with high AL. Moreover, in (SM stratified) analysis we were able to gain more granular insights into AL and SM and observed that gay/lesbian adults with high AL had a 4-fold increased risk of cancer death when compared to straight/heterosexual adults with low AL. Moreover, in fully adjusted models accounting for age, reported sex, race/ethnicity, income, educational attainment, smoking status, and alcohol consumption MSM and WSW had a 2-fold increased risk of cancer death.

This analysis provides empirical data in support of the minority stress model as described by Meyer et al., and this analysis takes the model beyond the subjective approaches to stress [28]. Stressors and mechanisms to cope with such stressors operate differently in oppressed population subgroups such as racial/ethnic minorities [29] and sexual minorities [28]. By considering biomarkers of stress, this analysis provides some objective evidence of the biological impact of social stressors. Indeed, subgroups with high AL whether gay or lesbian showed higher risks of cancer death, even after adjusting for age as there is growing evidence of higher AL in older adults [30]. Even after adjusting for sociodemographics and high-risk behaviors, these risks continue to be significantly associated with cancer death, thereby providing empirical evidence for the biological impact of minority stress on SMs. We found that gay and lesbian adults with high AL were at a nearly 5-fold age-adjusted increased risk of cancer death, while MSM and WSW with high AL had a 3-fold age-adjusted increased risk of cancer death. It is plausible that more social prejudice and biases are experienced by openly gay or lesbian individuals. Thus, gay and lesbian people are more likely to manifest minority stress (in the form of hyper-vigilance and resiliency) than those who report same sex behavior (i.e., MSM and WSW) but report heterosexual orientation.

There is considerable research indicating that SM individuals experience discrimination at higher rates than their heterosexual counterparts [31], which might result in psychological stress [32]. Studies specifically focused on cancer survivors show that SM survivors are impacted negatively with significant psychological distress and that their experience of discrimination was associated with both depression and anxiety, further impacting their ability to cope with cancer diagnosis [33]. Due to repeated encounters with discriminatory practices and limited competency of SM health within medicine, many SM may have a reduction in screening and preventive behaviors for cancer, thus placing them at higher risks for late-stage cancers [16,34]. Additionally, there is a greater risk of HPV-related anal cancer among cisgender sexual minority men and transgender women attributed to the limited cultural competency of health professionals regarding LGBTQIA+ care [35,36]. SM populations may experience inequities in health insurance coverage once diagnosed with cancer compared with their straight/heterosexual counterparts, due to discriminatory policies that did not recognize same-sex marriages prior to 2015 [37,38,39]. SM with high AL were less likely to be married or live with a partner when compared with straight/heterosexual individuals. Thus, it is possible cohabitation and marriage may serve as a proxy for social support, since SM with high AL experienced an increased risk of cancer mortality. Studies suggest improved survival and lower risks of mortality among cancer patients with higher social support [40,41,42,43]. 

The results of this study should be interpreted considering a few strengths and limitations. This is the first study to examine the association between sexual minorities, AL, and the risk of cancer mortality among a large representative sample of non-institutionalized US adults. We performed analysis using self-reported survey measures of sexual orientation; therefore, there are possible reporting biases for sexual orientation. Further, study participants may have given socially desirable responses based on their current relationship status, or identity or self-acceptance, and fear of stigmatization which may have underestimated the number of participants categorized as sexual minorities. US national surveys (e.g., NCHS, Center for Disease Control and Prevention) including NHANES are susceptible to information bias attributed to questionnaires with limited inclusivity for sexual and gender minorities. For example, during the baseline survey period (2001 through 2010) the NHANES questionnaire had limited answer choices for questions on sexual orientation (i.e., straight/heterosexual, gay/lesbian, bisexual, or something else) and gender identity (female or male at survey assessment). NHANES did not question on transgender and other gender identities. Thus, our categorization of SM status is only inclusive of a non-fluid, sexual orientation variable with a limited understanding of gender identity (and our rationale for not including “G” within SM). The possible misclassification of SM could underestimate the true association between sexual orientation and cancer mortality. Lastly, the measure of chronic physiologic stress, AL, was measured once at baseline when it is likely that AL changes over one’s life course. While our overall use of cancer death allowed us to examine several possible associations with SM and AL, the concatenation of our primary cancer mortality outcome does not allow for cancer-specific etiology. Moreover, patient-provider engagement and resources for healthcare may vary among sexual minority populations by geographic location driven by political and socio-cultural norms. Sexual minorities experience different access to healthcare based on rurality/urbanicity [44,45]. In our study, we did not examine the possible influence of geographic location, given that NHANES restricts the use of these data for public use. Future studies should explore the possible role of geographic location on sexual minority status with cancer mortality risks. Future studies should consider examining these associations with more granular measures of sexual orientation, gender identity, repeated measures of AL, and site-specific cancer mortality. Future research should also examine the intersection of race/ethnicity and SM, which may allow for understanding the complex and nuanced associations for racial minorities within gender and sexual orientation minority groups on risks for cancer mortality.

## 5. Conclusions

Findings from this study are foundational in re-energizing a research agenda focused on SM adults and health promotion, as has been supported by the National Academies of Science, Engineering and Medicine [46]. Given the lack of granularity for sexual minority status and the absence of gender identity, future studies and clinical endeavors should aim to be more inclusive regarding the collection of these data from sexual and gender minority populations. Furthermore, this study provides unequivocal evidence for the focus on cancer prevention, as supported by the growing calls by leading cancer organizations [47,48,49]. Finally, given the evidence from this study, there is an immediate need to focus on cancer prevention and control interventions that cultivate resiliency and address both subjective and objective stressors for SM adults; such interventions will need changing paradigms to incorporate inclusive participatory approaches so that SM individuals may contribute towards future research programs [50]. 

## Figures and Tables

**Figure 1 ijerph-20-06120-f001:**
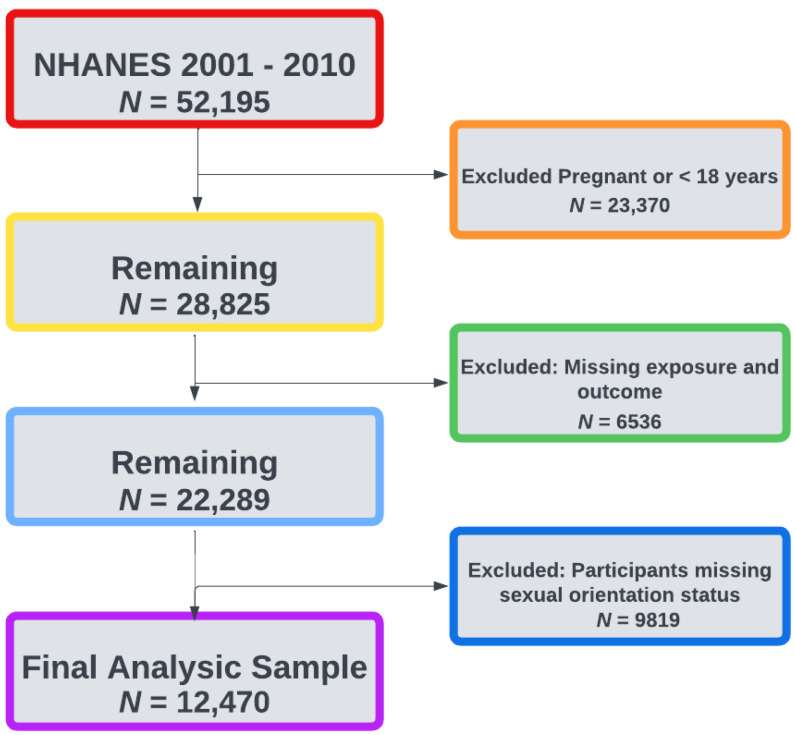
Flowchart for exclusion criteria for analytic sample of NHANES participants.

**Figure 2 ijerph-20-06120-f002:**
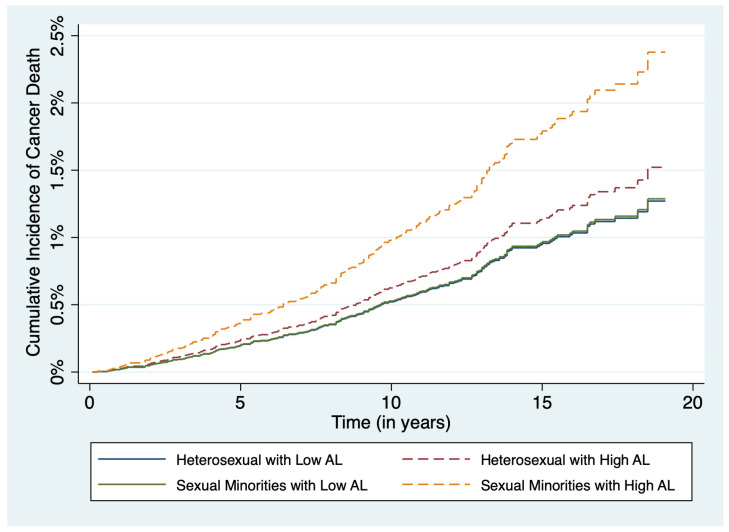
Unweighted age-adjusted cumulative incidence function competing risks plots for time to cancer death by sexual minority status and allostatic load status. Sexual Minority groups include, gay, lesbian, (MSM) men who have sex with men and (WSW) women who have sex with women. MSM and WSW are participants in the absence of a current lesbian, gay or bisexual identity.

**Table 1 ijerph-20-06120-t001:** Socio-demographic characteristics, personal health, and medical conditions by allostatic load and sexual orientation, National Health Examination Survey (NHANES) study period. Among 12,470 NHANES survey participants (an estimated 46,347,598 non-institutionalized US adults) years 2001 through 2010 and follow-up through 31 December 2019.

	Living with High Allostatic Load ^a^				Living with Low Allostatic Load			
	Gay/Lesbian	Bisexual	MSM and WSW ^c^	Straight/ Heterosexual	Gay/Lesbian	Bisexual	MSM and WSW ^c^	Straight/ Heterosexual
**Unweighted sample size ^b^**	64	112	150	4957	128	177	208	6674
**Weighted sample size ^b^**	664,265	1,049,553	1,449,998	46,347,598	1,395,728	1,585,265	2,116,731	68,120,312
	**Presented as % or Mean with** **SE ^d^**							
**Allostatic load total score ^e^**	4.1 (0.2)	4.3 (0.2)	4.1 (0.1)	4.1 (0.02)	1.0 (0.07)	0.9 (0.05)	1.0 (0.05)	1.0 (0.01)
**Reported female sex at survey**	38.3 (8.0)	68.9 (5.2)	72.0 (3.3)	48.4 (0.7)	35.8 (5.8)	69.4 (3.9)	67.3 (3.7)	46.7 (0.6)
**Mean age in years**	42.4 (1.2)	38.7 (1.3)	43.5 (1.0)	43.9 (0.2)	37.5 (1.0)	32.1 (1.0)	36.0 (0.9)	36.5 (0.2)
**Age Group**								
18–29	11.9 (4.0)	24.2 (4.7)	8.8 (3.1)	11.0 (0.5)	24.4 (3.9)	46.7 (4.8)	34.6 (3.6)	33.2 (0.8)
30–39	20.3 (5.3)	26.6 (5.5)	19.7 (3.8)	20.8 (0.7)	35.7 (4.9)	30.4 (4.1)	29.6 (3.6)	26.1 (0.7)
40–49	45.4 (8.6)	31.7 (4.8)	39.9 (5.0)	33.1 (0.8)	25.7 (5.6)	13.2 (2.9)	19.0 (3.3)	24.9 (0.7)
50–59	22.5 (6.6)	17.4 (4.6)	31.5 (5.5)	35.1 (1.0)	14.1 (3.9)	9.8 (2.7)	16.8 (3.2)	15.8 (0.6)
**Race/Ethnicity**								
Non-Hispanic White	69.3 (5.9)	70.9 (4.3)	70.7 (3.8)	68.4 (1.8)	76.2 (3.9)	72.3 (3.6)	74.5 (3.7)	72.3 (1.3)
Non-Hispanic Black	12.5 (3.7)	14.5 (2.9)	14.9 (3.0)	14.3 (1.0)	7.5 (1.8)	13.6 (2.4)	7.8 (1.7)	7.9 (0.6)
Hispanic	8.1 (2.6)	10.9 (2.9)	9.9 (2.1)	12.7 (1.3)	11.0 (2.5)	11.4 (2.4)	10.9 (2.0)	14.4 (1.0)
Other and Mixed Race	10.1 (4.1)	3.7 (2.0)	4.5 (1.9)	4.6 (0.4)	5.2 (2.0)	2.6 (1.3)	6.8 (1.9)	5.4 (0.4)
**Married or lives with partner**	28.2 (7.0)	39.1 (5.9)	54.9 (5.0)	70.0 (1.1)	30.8 (6.7)	42.9 (4.2)	60.3 (3.9)	64.8 (0.94)
**Education**								
<High school	12.8 (5.3)	17.6 (4.3)	14.0 (2.4)	17.3 (0.7)	5.3 (1.7)	20.9 (3.5)	10.2 (2.3)	14.2 (0.8)
High school/GED	12.0 (5.5)	23.1 (5.2)	18.6 (4.8)	26.7 (0.9)	13.2 (3.1)	21.7 (3.7)	17.2 (3.3)	23.8 (0.7)
Some coll. or assoc. degree	41.4 (6.8)	39.6 (6.0)	42.8 (5.1)	33.2 (1.0)	27.8 (5.8)	35.4 (4.0)	46.6 (4.8)	31.7 (0.7)
College graduate	33.8 (5.8)	19.7 (4.7)	24.6 (4.5)	22.7 (1.0)	53.7 (6.6)	22.1 (4.5)	24.1 (3.7)	30.3 (1.1)
Missing	0 (0.0)	0 (0.0)	0 (0.0)	0.04 (0.03)	0 (0.0)	0 (0.0)	0 (0.0)	0.1 (0.04)
**<$20,000 annual household income**	17.7 (4.9)	31.1 (5.5)	18.0 (4.1)	13.8 (0.7)	16.5 (4.1)	26.7 (3.8)	20.0 (3.2)	12.8 (0.6)
**Has health insurance**	83.0 (4.5)	67.1 (5.2)	75.8 (4.1)	80.0 (0.8)	79.4 (3.6)	66.7 (4.6)	77.4 (2.8)	76.3 (0.8)
**Has a routine place to go for healthcare**	86.2 (5.8)	88.8 (3.4)	88.4 (2.9)	86.1 (0.6)	83.2 (4.1)	79.7 (3.8)	79.7 (3.0)	80.9 (0.7)
**Visits clinics/doctors’ office ^f^**	84.6 (5.8)	80.9 (4.2)	81.0 (3.4)	81.5 (0.6)	76.6 (4.4)	72.2 (4.1)	73.0 (3.4)	76.0 (0.9)
**1 or more healthcare visits a year ^g^**	80.5 (5.5)	90.4 (3.1)	87.7 (2.9)	83.1 (0.7)	85.0 (3.6)	88.9 (2.3)	83.1 (2.8)	78.8 (0.7)
**Current smoker status**	34.8 (6.9)	39.5 (5.3)	41.2 (5.7)	26.7 (0.9)	32.8 (5.5)	45.2 (4.6)	36.0 (3.9)	26.0 (0.8)
**Current Heavy Drinker ^h^**	38.8 (5.8)	63.5 (5.0)	41.8 (5.2)	39.6 (1.0)	42.0 (6.1)	61.4 (4.8)	53.1 (3.8)	46.1 (1.0)
**Any cancer history ^i^**	5.2 (3.9)	2.2 (1.4)	9.6 (2.9)	6.1 (0.4)	2.0 (1.6)	6.5 (2.3)	2.0 (1.0)	4.0 (0.4)
**Ever heart attack**	2.6 (2.6)	2.1 (1.2)	2.4 (1.4)	2.5 (0.3)	2.0 (1.5)	1.7 (1.1)	0 (0.0)	0.8 (0.1)
**Ever STD ^j^**	7.2 (2.9)	14.0 (3.6)	16.3 (3.7)	7.3 (0.5)	9.8 (3.2)	16.0 (3.4)	21.2 (3.1)	8.3 (0.4)
**HIV positive**	4.9 (3.0)	2.7 (1.3)	0 (0.0)	0.2 (0.1)	8.5 (2.6)	1.1 (0.6)	0 (0.0)	0.1 (0.02)
**Depressive disorder**	6.8 (2.9)	22.3 (5.7)	17.4 (3.6)	8.8 (0.7)	9.2 (3.3)	19.3 (3.8)	15.5 (3.0)	5.4 (04)

^a^ High Allostatic load is defined as total Allostatic load score greater than or equal to 3 (presented as column percentages and standard errors). ^b^ Unweighted sample size. ^c^ (MSM) men who have sex with men and (WSW) women who have sex with women are participants in the absence of a current lesbian, gay or bisexual identity. ^d^ Estimated using sampling weights from National Health and Nutrition Examination Survey (NHANES). ^e^ Presented as weighted column proportion (standard error) or mean (standard error) for continuous variables. ^f^ Defined as self-reported response to what kind/type of place most often go to for healthcare. ^g^ Defined as self-reported response to number of times received healthcare over the past year. ^h^ Heavy drinker is defined as having 2 or more drinks a day for reported female at survey, and 3 or more drinks a day for reported male at survey. ^i^ Defined as self-reported response to ever being diagnosed by a doctor or health professional of any cancer or malignancy. ^j^ Defined as self-reported response to ever being diagnosed with genital herpes, genital warts, gonorrhea, and chlamydia.

**Table 2 ijerph-20-06120-t002:** Survey weighted Cox proportional hazard models presented as Hazard Ratios (HR) and 95% Confidence Intervals (CI) for the association between sexual orientation/allostatic load and risk of cancer death, among 12,470 (weighted *n* = 122,729,451) NHANES participants with 222 (weighted *n* = 1,955,041) cancer-related deaths.

	No. of Participants (Weighted N)	No. of Cancer Deaths (Weighted%)	Mean Survival Months (*SE*)	Hazard Ratio (HR) and 95% Confidence Interval (*CI*)	
				Model 1	Model 2
**Sexual Orientation and Allostatic Load Status**					
Straight/heterosexual with low allostatic load	6674 (68,120,312)	78 (1.2)	207.6 (0.2)	1.00 (Referent)	1.00 (Referent)
Straight/heterosexual with high allostatic load	4957 (46,347,598)	128 (2.6)	218.3 (0.3)	1.22 (0.81–1.84)	1.06 (0.68–1.67)
Gay, Lesbian, Bisexual, MSM, WSW with low allostatic load	513 (5,097,724)	5 (1.0)	118.4 (0.4)	0.63 (0.24–1.66)	0.57 (0.22–1.48)
Gay, Lesbian, Bisexual, MSM, WSW with high allostatic load	326 (3,163,817)	11 (3.4)	162.6 (0.9)	3.32 (1.86–5.93)	2.55 (1.40–4.65)
**Among those living with high allostatic load**					
Straight/heterosexual	4957 (46,347,598)	128 (2.6)	218.3 (0.3)	1.00 (Referent)	1.00 (Referent)
Gay, Lesbian, Bisexual, MSM, WSW	326 (3,163,817)	11 (3.4)	162.6 (0.9)	2.66 (1.54–4.61)	2.26 (1.33–3.84)
**Among those living with low allostatic load**					
Straight/heterosexual	6674 (68,120,312)	78 (1.2)	207.6 (0.2)	1.00 (Referent)	1.00 (Referent)
Gay, Lesbian, Bisexual, MSM, WSW	513 (5,097,724)	5 (1.0)	118.4 (0.4)	0.64 (0.24–1.69)	0.58 (0.22–1.54)
*p*-value for interaction between sexual orientation and allostatic load				0.0076	0.008

Percentages are weighted. Cox proportional hazard models are estimated using NHANES survey weighting. (MSM) men who have sex with men and (WSW) women who have sex with women are participants in the absence of a current lesbian, gay or bisexual identity. Model 1 is adjusted for age, Model 2 is adjusted for age, and sociodemographic factors including reported sex at survey, race, income, education, smoking status, and alcohol consumption.

## Data Availability

Data for this study participants are available from National Health and Nutrition Examination Survey at https://wwwn.cdc.gov/nchs/nhanes/. Data on National Death Index mortality linked files are available from the National Center for Health Statistics at https://www.cdc.gov/nchs/ndi/index.htm (accessed on 1 June 2022).

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
