# Peer review of "Investigating the Joint Effect of Allostatic Load among Lesbian, Gay, and Bisexual Adults with Risk of Cancer Mortality"

_ijerph, 2023, doi:10.3390/ijerph20126120_

Round 1
Reviewer 1 Report
This is an interesting secondary analysis of NHANES data, looking at risk of cancer mortality based on sexual minority (SM) status and allostatic load (AL) level. Overall the manuscript would be of interest to readers. A few suggestions for improvement:
1. p 2 ln 54, While the background on cancer screening is important, it feels like a nonsequitur. Consider framing this statement in the context of a framework like minority stress that can explain both AL and healthcare engagement.
2. p 2 ln 60, More background literature on the link between AL and cancer mortality is needed. Present the increase in risk of mortality among heterosexuals with AL from prior research.
3. p 4 ln 110, State in the methods that gender identity was not assessed in NHANES, and so it is impossible to know if there are trans persons in the sample. I recommend moving away from "sex as a biological variable" here; describe how the sex variable was assessed in NHANES and how you're using it, rather than referencing biology.
4. p 4 ln 115, given that cutoffs for high AL have been determined in prior literature, why was an idiosyncratic 75%+ cutoff chosen for this study, rather than using a prior cut point?
5. p 4 ln 125, consider "categorized into eight groups," rather than talking about "levels."
6. p 4 ln 142, were Asian and American Indian races not assessed? If so, why were they combined into "Other?"
7. p 5 ln 195, what was the difference in AL level by SM status? Did SM people have higher AL (continuous), or were they more likely to be categorized as high AL (dichotomous)?
8. p 5 ln 205, the comparison between SM with high AL and heterosexuals with low AL on demographic characteristics is confusing, as two factors are being varied. Consider instead comparing the high AL groups, as we are most concerned about their cancer mortality.
9. p 8 ln 242, as there were only 5 cancer related deaths among SM people with low AL overall, I am unsure how the cumulative incidence graph was calculated for the smaller subgroups, like bisexuals and MSM/WSW. Were there even any cancer deaths observed? If not, consider presenting a curve for all SM with low AL, and all SM with high AL.
10. p 8 ln 242, for colorblind and other readers, consider using different line types (dashes, dots, etc.) to differentiate the graph.
11. p 10 ln 317, the sentence about marriage and social support does not make sense as written. Please revise.
Author Response
We’d like to thank the International Journal of Environmental Research and Public Health editors and reviewers for your time re-reviewing our manuscript and the thoughtful and thorough comments provided. We have carefully reviewed the comments and have addressed each concern point-by-point below. We have uploaded these documents to the International Journal of Environmental Research and Public Health resubmission portal per the instructions provided in your email.
RESPONSES TO COMMENTS- Manuscript ID: ijerph-2280616
Reviewer 1 Comments:
This is an interesting secondary analysis of NHANES data, looking at risk of cancer mortality based on sexual minority (SM) status and allostatic load (AL) level. Overall, the manuscript would be of interest to readers. A few suggestions for improvement:
- p 2 ln 54, While the background on cancer screening is important, it feels like a no sequitur. Consider framing this statement in the context of a framework like minority stress that can explain both AL and healthcare engagement.
- We thank the reviewer for this comment and have provided more framework on how minority stress can explain both allostatic load and healthcare engagement.
- p 2 ln 60, More background literature on the link between AL and cancer mortality is needed. Present the increase in risk of mortality among heterosexuals with AL from prior research.
- We thank the reviewer for this comment. While there has been literature on AL and risk of cancer morality, as well as mortality from other diseases, these analysis have not been stratified examined by sexual or gender minority status and therefore we would be able to delineate the risk for just heterosexual adults.
- p 4 ln 110, State in the methods that gender identity was not assessed in NHANES, and so it is impossible to know if there are trans persons in the sample. I recommend moving away from "sex as a biological variable" here; describe how the sex variable was assessed in NHANES and how you're using it, rather than referencing biology.
- We thank the reviewer for this suggestion and have modified the methods to show NHANES assessed gender in questionnaire.
- p 4 ln 115, given that cutoffs for high AL have been determined in prior literature, why was an idiosyncratic 75%+ cutoff chosen for this study, rather than using a prior cut point?
- We thank the reviewer for this comment. Our previous study examined allostatic load among a larger population size, while the current analysis population is smaller (n= 12,470). We kept our definition of allostatic load components consistent with prior literature for thresholds of each biomarker components. We examined allostatic load components among our population and kept the same percentile cut off points (25% and 75%) as previous literature.
- p 4 ln 125, consider "categorized into eight groups," rather than talking about "levels."
- We thank the reviewer for this comment and have modified to “categorized into eight groups”.
- p 4 ln 142, were Asian and American Indian races not assessed? If so, why were they combined into "Other?"
- We thank the reviewer for this comment. Asian and American Indian races we not specifically assessed in NHANES recode variable.
- p 5 ln 195, what was the difference in AL level by SM status? Did SM people have higher AL (continuous), or were they more likely to be categorized as high AL (dichotomous)?
- We thank the reviewer for their comment, in the results section we have included continuous allostatic load scores among the 4 sexual minorities groups (AL mean scores: gay/lesbian = 1.66, SE = 0.12; bisexual = 1.92, SE = 0.12; homosexually experienced = 1.85, SE = 0.12; heterosexual/straight = 1.90, SE = 0.03).
- p 5 ln 205, the comparison between SM with high AL and heterosexuals with low AL on demographic characteristics is confusing, as two factors are being varied. Consider instead comparing the high AL groups, as we are most concerned about their cancer mortality.
- We thank the reviewer for their concern, however, our a priori objective were to examine the interaction between high allostatic load with sexual minority status on cancer mortality. Thus, we elected to examine whether groups of sexual minority status with high allostatic load were different from those with sexual minority status with low allostatic load (referent group = heterosexuals with low allostatic load).
- p 8 ln 242, as there were only 5 cancer related deaths among SM people with low AL overall, I am unsure how the cumulative incidence graph was calculated for the smaller subgroups, like bisexuals and MSM/WSW. Were there even any cancer deaths observed? If not, consider presenting a curve for all SM with low AL, and all SM with high AL.
- We thank the reviewer or this comment. We have added the original Figure 2 (stratified by SM status) to the supplementary material, as it is representative of our supplementary table (Supplementary Table 1). Additionally, we have modified our table 2 to be among all SM with High AL and SM with Low AL vs Heterosexual with High AL and Heterosexual with Low AL.
- p 8 ln 242, for colorblind and other readers, consider using different line types (dashes, dots, etc.) to differentiate the graph.
- We thank the reviewer for this comment and have incorporated dashed lines to those with high AL in Figure 2.
- p 10 ln 317, the sentence about marriage and social support does not make sense as written. Please revise.
- We thank the reviewer for this suggestion and have revised the sentence accordingly.
Reviewer 2 Report
• The introduction does an appropriate job of framing the paper and
inviting the reader to continue.
• I found these sentences particularly on-target: Lines
59--64 and Lines 354-358. These are key points and make this work both timely and important.
• The data utilized looked at key biomarkers but does not seem to take into account patient characteristics/behaviors like drinking or smoking, known to be important in cancer risk assessment. If this data from NHANES is not used, explaining why such characteristics were not included would be useful and should be discussed.
• The conclusion does reflect what was studied and was succinct. However, it missed the opportunity to discuss that NHANES data can be accessed that provides some geographic information that may be important in the analysis of cancer mortality risk, which may also impact both exposures to carcinogens but also to the potential of underreporting of sexual orientation (e.g., rural vs. urban or CA vs. Wyoming). Further analyses could recognize that such locational biases may impact patient-doctor relationships and thus could allow physicians to alter their patient interactions accordingly
Author Response
We’d like to thank the International Journal of Environmental Research and Public Health editors and reviewers for your time re-reviewing our manuscript and the thoughtful and thorough comments provided. We have carefully reviewed the comments and have addressed each concern point-by-point below. We have uploaded these documents to the International Journal of Environmental Research and Public Health resubmission portal per the instructions provided in your email.
RESPONSES TO COMMENTS- Manuscript ID: ijerph-2280616
Reviewer 2 Comments:
Suggestions for Authors
- The introduction does an appropriate job of framing the paper and inviting the reader to continue.
- Many thanks for the kind words!
- I found these sentences particularly on-target: Lines 59--64 and Lines 354-358. These are key points and make this work both timely and important.
- Thank you!
- The data utilized looked at key biomarkers but does not seem to take into account patient characteristics/behaviors like drinking or smoking, known to be important in cancer risk assessment. If this data from NHANES is not used, explaining why such characteristics were not included would be useful and should be discussed.
- We thank the reviewer for this comment and have updated our Model 2 to include accounting for alcohol consumption and tobacco use. This has additionally been explained in the methods.
- The conclusion does reflect what was studied and was succinct. However, it missed the opportunity to discuss that NHANES data can be accessed that provides some geographic information that may be important in the analysis of cancer mortality risk, which may also impact both exposures to carcinogens but also to the potential of underreporting of sexual orientation (e.g., rural vs. urban or CA vs. Wyoming). Further analyses could recognize that such locational biases may impact patient-doctor relationships and thus could allow physicians to alter their patient interactions accordingly.
- We thank the reviewer for this suggestion and have incorporated suggestions into the discussion.
Reviewer 3 Report
Abstract:
1. Line 20: “which may be explained by consistent experiences of discriminatory practices”, I suggest rephrasing this sentence to denote that discrimination is not the only factor. When read, it seems to suggest cause and effect.
2. Line 20: “This is the first study to examine…” I suggest to not make that statement. Maybe one of the first? Feeling the gap? Absolutes increase the probability of being wrong.
3. Line 27: the use of “straight” could be problematic and pejorative. Please use only heterosexual, unless straight/heterosexual term was the one used on the survey. If this true, please justified it later on the manuscript that for clarification.
Introduction:
4. Line 62: I suggest changing homophobia for “homophobia/biphobia” since bisexual individuals are part of the study.
5. Line 69: There is an error in the beginning of the sentence “NHANES is can be linked with…”
6. Figure 1 looks pixelated.
7. Line 88. IRB place and identification number should be provided for corroboration and transparency.
8. Line 89-102. Please explain if the heterosexual group was for comparation purpose. I was a little confused since the topic is Sexual Minority Status.
9. Line 111: citation error.
10. Statistic symbols such n’s should be in italic
Author Response
We’d like to thank the International Journal of Environmental Research and Public Health editors and reviewers for your time re-reviewing our manuscript and the thoughtful and thorough comments provided. We have carefully reviewed the comments and have addressed each concern point-by-point below. We have uploaded these documents to the International Journal of Environmental Research and Public Health resubmission portal per the instructions provided in your email.
RESPONSES TO COMMENTS- Manuscript ID: ijerph-2280616
Reviewer 3 Comments:
Abstract:
- Line 20: “which may be explained by consistent experiences of discriminatory practices”, I suggest rephrasing this sentence to denote that discrimination is not the only factor. When read, it seems to suggest cause and effect.
- We thank the reviewer for their comment and have modified the sentence.
- Line 20: “This is the first study to examine…” I suggest to not make that statement. Maybe one of the first? Feeling the gap? Absolutes increase the probability of being wrong.
- We thank the reviewer for their comment, and have modified the sentence accordingly.
- Line 27: the use of “straight” could be problematic and pejorative. Please use only heterosexual, unless straight/heterosexual term was the one used on the survey. If this true, please justified it later on the manuscript that for clarification.
- Many thanks to the reviewer for this comment, this is terminology used by NHANES investigators. We have modified the methods to demonstrate this.
Introduction:
- Line 62: I suggest changing homophobia for “homophobia/biphobia” since bisexual individuals are part of the study.
- Thank you for this suggestion, it has been included.
- Line 69: There is an error in the beginning of the sentence “NHANES is can be linked with…”
- Thank you for catching this typographical error, we have modified accordingly.
- Figure 1 looks pixelated.
- We thank the reviewer for this feedback and have uploaded a new variation of the image.
- Line 88. IRB place and identification number should be provided for corroboration and transparency.
- We thank the reviewer for this comment. At Augusta University this study falls under exempt category number 4, and does not require IRB approval due to the use of publicly available., de-identified information. Please see more information: https://augustauniversity.app.box.com/s/kc1rna1vjaebame5u6fnuh6neinky9c3
- Line 89-102. Please explain if the heterosexual group was for comparation purpose. I was a little confused since the topic is Sexual Minority Status.
- We thank the reviewer for this comment and have clarified that heterosexual was used as a reference group for statistical analysis.
- Line 111: citation error.
- Thank you for catching this error we have updated citation to correct formatting.
- Statistic symbols such n’s should be in italic
- We thank the reviewer for this comment and have italicized statistic symbols.